

# Genetic diversity and structure of the critically endangered *Artocarpus annulatus,* a crop wild relative of jackfruit (*A. heterophyllus*)

Leta Dickinson[1,*],  Hilary Noble[2],  Elliot Gardner[3,4,5],
Aida Shafreena Ahmad Puad[6],  Wan Nuur Fatiha Wan Zakaria[6] and
Nyree J.C. Zerega[1,2,*]

[1] Plant Biology and Conservation, Northwestern University, Evanston, IL, United States of America
[2] The Negaunee Institute for Plant Conservation Science and Action, Chicago Botanic Garden, Glencoe, IL, United States of America
[3] Department of Biology, Case Western Reserve University, Cleveland, OH, United States of America
[4] The Morton Arboretum, Lisle, IL, United States of America
[5] Singapore Botanic Gardens, Singapore, Singapore
[6] Faculty of Resource Science and Technology, Universiti Malaysia Sarawak, Kota Samarahan, Sarawak, Malaysia
[*] These authors contributed equally to this work.

Corresponding author
Nyree J.C. Zerega, n-zerega@northwestern.edu

## ABSTRACT

Limestone karsts of Southeast Asia can harbor high levels of endemism, but are highly fragmented, increasingly threatened, and their biodiversity is often poorly studied. This is true of the Padawan Limestone Area of Sarawak, Malaysia, home to the endemic *Artocarpus annulatus,* the closest known wild relative of two important and underutilized fruit tree crops, jackfruit (*A. heterophyllus*) and cempedak (*A. integer*). Identifying and conserving crop wild relatives is critical for the conservation of crop genetic diversity and breeding. In 2016 and 2017, five *A. annulatus* populations were located, and leaf material, locality information, and demographic data were collected. Microsatellite markers were used to assess genetic diversity and structure among populations, and to compare levels of genetic diversity to closely related congeneric species. Results indicate no evidence of inbreeding in *A. annulatus*, and there is no genetic structure among the five populations. However, diversity measures trended lower in seedlings compared to mature trees, suggesting allelic diversity may be under threat in the youngest generation of plants. Also, genetic diversity is lower in *A. annulatus* compared to closely related congeners. The present study provides a baseline estimate of *A. annulatus* genetic diversity that can be used for comparison in future studies and to other species in the unique limestone karst ecosystems. Considerations for in situ and ex situ conservation approaches are discussed.

## INTRODUCTION

The karst limestone formations of Southeast Asia are dramatic, rugged landscapes that are home to some of the most unique and understudied flora and fauna in the world (*Tuyet, 2001*; *Clements et al., 2006*; *Chung et al., 2014*; *Hughes, 2017*). There are approximately 800,000 km$^2$ of karst ecosystems in tropical Southeast Asia and southern China, and they include several UNESCO World Heritage sites (*Day & Urich, 2000*; *Williams, 2008*). Most karsts were formed millions of years ago by calcium-secreting marine organisms, and the current landscapes are the result of geomorphological processes that have formed complex terrains such as fissured and precipitous cliffs, jagged towers, sinkholes, extensive cave systems, and subterranean rivers, harboring a range of ecological niches (*Clements et al., 2006*; *Williams, 2008*). The resulting landscapes are naturally fragmented mosaics of karst and ordinary terrain, promoting edaphic isolation and high levels of endemism (*Kruckeberg & Rabinowitz, 1985*; *Farjon et al., 2002*; *Schilthuizen, 2004*; *Musser et al., 2005*; *Williams, 2008*). Healthy karst habitats provide tangible benefits to humans, such as water from below-ground aquifers, as well as a range of ecosystem services (*Brinkmann & Garren, 2011*), some of which are due to their unique biodiversity (*Schilthuizen, 2004*; *Clements et al., 2006*). However, due to human practices, the karst ecosystems and the species inhabiting them are themselves increasingly at risk of further fragmentation and degradation (*Clements et al., 2006*; *Clements et al., 2008*; *Struebig et al., 2009*; *Latinne, Waengsothorn & Michaux, 2011*; *Hughes, 2017*). The most immediate and direct threat is overexploitation of resource extraction, especially limestone and mineral mining (*Kiew et al., 2017*; *Brinkmann & Garren, 2011*; *Hughes, 2017*). These extremely lucrative but destructive extractive practices can have cascading effects of severe fragmentation and habitat degradation, leading to serious negative impacts on flora, fauna, and humans.

In Sarawak, Malaysia, the Padawan limestone formation (in the southwest part of Kuching Division) is the largest and one of the oldest (dating from about 163 to 100 million years ago) outcrops among the six limestone biodistricts there (*Lim, 2008*; *Cranbrook, 2004*). One of the species endemic to this habitat is the critically endangered tree, *Artocarpus annulatus* Jarrett, known only from the Padawan-Serian-Tebedu karst forests (Fig. 1). This species is a member of the mulberry (Moraceae) family and the closest wild relative of the important tropical food crops jackfruit (*A. heterophyllus* Lam.) and cempedak (*A. integer* (Thunb.) Merr.) (*Zerega, Nur Supardi & Motley, 2010*; *Williams et al., 2017*; *Zerega et al., 2019*). *Artocarpus annulatus* was first described nearly 45 years ago (*Jarrett, 1975*) but remains poorly understood in terms of its biology and distribution. As of 2015, the species was known from only nine herbarium specimens. They were all collected between 1960–1999 and came from just four limestone hill forest outcrops within the Padawan-Serian-Tebedu limestone karst formation in Kuching (Gunung Mentawa, Gunung Gayu, and Gunung Teng Bukap) and Serian (Gunung Payang) Divisions (Fig. 2). According to *Zerega et al. (2019)*, there are thought to be fewer than 50 mature individuals in an area of occupancy (AOO) of 32 km$^2$.

The importance of studying and conserving *A. annulatus* has broad implications beyond karst systems, as the genetic diversity of crop wild relatives can prove a useful

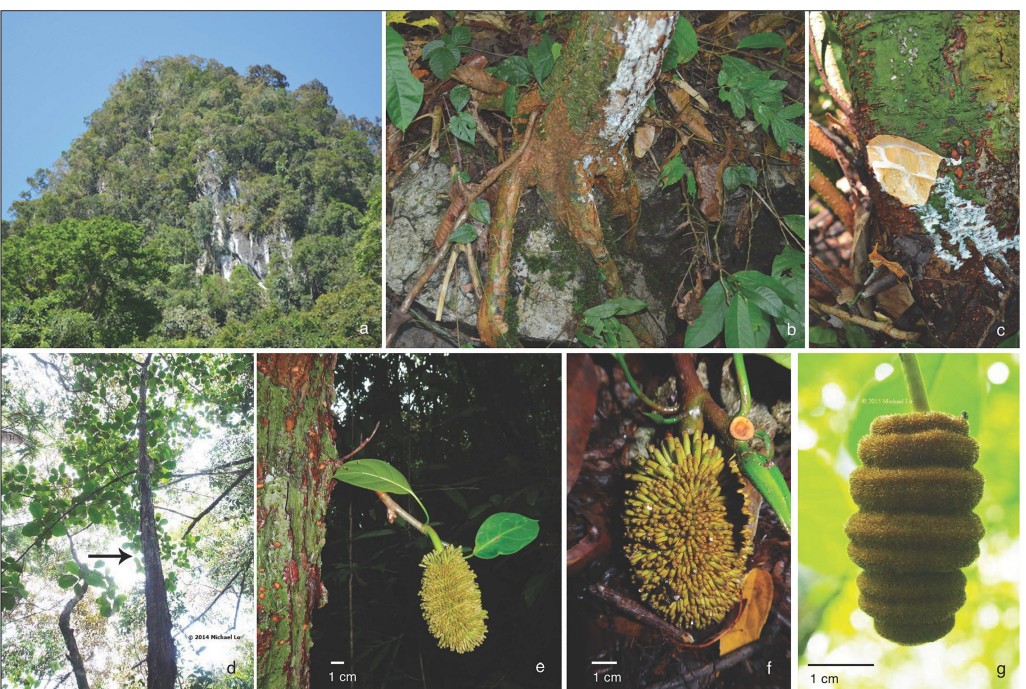

**Figure 1** **Artocarpus annulatus.** (A) Karst limestone habitat in Kuching Division, Sarawak, (B) Roots growing on limestone outcrops, (C) White exudate found throughout trees, (D) Tree, (E) Cauliflorous syncarp (multiple fruit structure), (F) Close up of syncarp, and (G) Male inflorescence with annulate rings. Photo credits: Nyree Zerega (A–C), Michael Lo (D and G) and Benedikt Kuhnhäuser (E and F).

source of novel traits (*Lobell et al., 2008*). The economic and food value of its cultivated close relatives, cempedak and jackfruit, are likely to increase, as they grow in tropical regions where food insecurity is high, and provide greater local autonomy in food production systems (*Ford-Lloyd et al., 2011*; *Jones et al., 2011*; *Wang et al., 2018*; *Witherup et al., 2019*). As agriculture adapts to climate change, long-lived tree crops may play an increasingly important role in the global diet. It is known that jackfruit and cempedak are interfertile, because hybrid cultivars exist (*Wang et al., 2018*). Given that *A. annulatus* is the only other member of the lineage including jackfruit and cempedak (*Williams et al., 2017*), it is possible that they could all be interfertile and traditional plant breeding could incorporate desirable traits, such as expanding the edaphic conditions of the crop species.

Depending on what we know about the genetic diversity and structure of endangered species, different approaches to conservation may be warranted (*Widener & Fant, 2018*; *Münzbergová, 2005*; *Crozier, 1997*). Due to the unique karst environment, it is possible that *A. annulatus* populations have long been isolated and small. However, human-caused fragmentation may have further reduced genetic diversity, making it harder for the species to rebound and recover (*Leimu et al., 2006*). While there are limited studies examining population genetics of endangered flora in limestone karst systems of Southeast Asia, the unique habitat and fragmented nature of karst systems have made them of interest for understanding species diversification through isolation. Recent studies of common

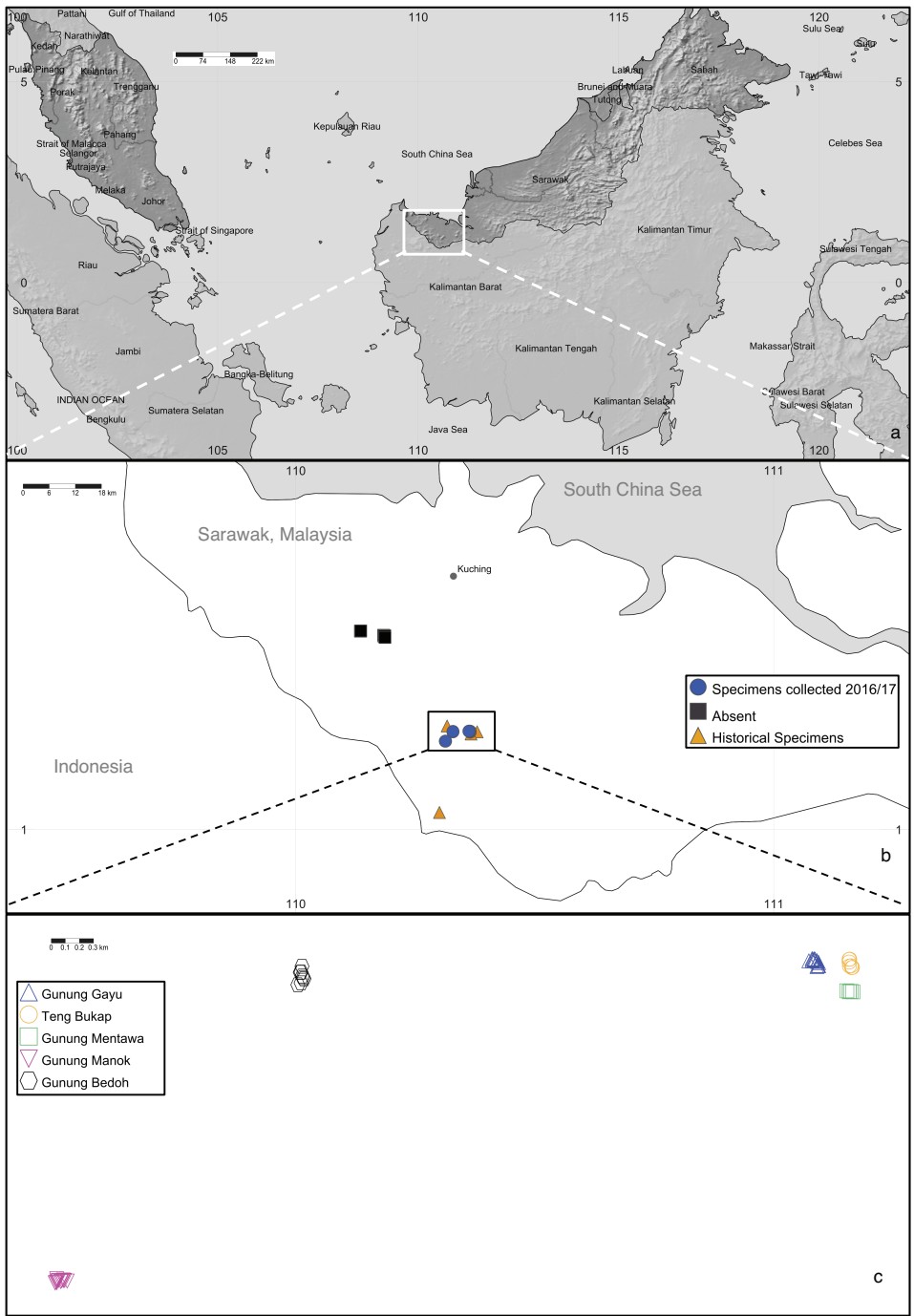

**Figure 2** *Artocarpus annulatus* **distribution.** (A) Malaysia is shown in dark gray; the area in Sarawak where *A. annulatus* populations are found is indicated by the white box, (B) close up of boxed area from above panel shows extant populations of *A. annulatus* (circles), locations of historical specimens (triangles), and locations that were searched in 2016 and no *A. annulatus* were found (squares), (C) close up of boxed area from above panel shows known extant *A. annulatus* populations that were sampled in this study. Map Credit: SimpleMappr (2019), CC 0.

karst endemic plant genera (*Begonia* and *Primulina*) found substantial differentiation and significant isolation by distance across populations (*Gao et al., 2015*; *Tseng et al., 2019*). Understanding genetic patterns and diversity in *A. annulatus* is an important step towards developing conservation strategies for this crop wild relative. This study uses microsatellite and demographic data to assess the genetic diversity and structure of *A. annulatus,* an endemic tree of the Padawan-Serian-Tebedu karst forests of Sarawak, Malaysia. Comparisons of genetic diversity are also made to its two closest relatives, which are commonly occurring underutilized crops. Specifically, we test the following hypotheses: Given the relatively recent (in decades) destruction of *A. annulatus* habitat, the known populations of *A. annulatus* will show recent evidence of gene flow and as there will not be population specific alleles; due to pressures of increased fragmentation and habitat loss, recruitment of younger individuals may be more tenuous and lead to decreased genetic diversity in younger individuals compared to mature trees; and as a critically endangered species with restricted distribution, *A. annulatus* will harbor less genetic diversity than its widely distributed congeners. It is hoped that the results can inform management plans for in situ and ex situ conservation efforts and provide much needed information about the unique biota of the Southeast Asian karst formations. Understanding how diversity is distributed among populations will provide insight into the importance of whether in situ or ex situ conservation efforts should be targeted toward all or some populations to capture the existing diversity of this critically endangered species.

## MATERIALS & METHODS

### Species information

*Artocarpus annulatus* is a small to mid-sized evergreen tree, growing up to 10 m tall on karst outcrops in Sarawak, Malaysia. It is monoecious, with separate male and female inflorescences occurring on the same individual tree. The male inflorescences have rings around them, for which the specific epithet "*annulatus*" is named (Fig. 1). Once fertilized, the female inflorescence develops into a multiple fruit structure (called a syncarp) that is cauliflorous (i.e., grows directly from the tree trunk) (Fig. 1). Like all members of the genus *Artocarpus,* it produces copious white exudate from all parts when cut. The closest relatives of *A. annulatus* are two important crop species: *A. heterophyllus* (jackfruit) and *A. integer* (cempedak). They also bear cauliflorous fruit structures, which are edible and much larger than *A. annulatus* (reaching up to 45 kg in *A. heterophyllus*). Jackfruit is thought to be native to the western Ghats of India (with a possible secondary center of diversity in Bangladesh) (*Witherup et al., 2019*). It is grown throughout much of the tropics today. Cempedak (*A. integer* var. *integer*) is thought to be native to peninsular Malaysia and possibly Borneo; bangkong (*A. integer* var. *silvestris*) is a close wild relative native to peninsular Malaysia (*Wang et al., 2018*).

### Site information

*Artocarpus annulatus* is known only from the Padawan limestone area in Kuching District, Sarawak, Malaysia (Fig. 2). This area consists of small mountains with tower karst formations. The landscape is dominated by very steep, or even vertical sided

limestone "towers", known as tower karst formations, which are the result of once continuous limestone rock being eroded (*Brenda, Gendang & Ambun, 2004*). The towers are sculptured by chemical and mechanical processes forming a terrain covered by jagged pinnacles and deep pits and crevices. *Artocarpus annulatus* can be found at the base of the karst forest mountains on relatively flat ground, as well as at higher elevations among the karst towers and growing sideways from outcrops, with roots clinging to the limetone rock (Fig. 1).

## Sampling

In 2016 and 2017, fieldwork was undertaken in the Padawan-Serian-Tebedu karst limestone forests in Sarawak, Malaysia for collection and observation of *A. annulatus* (*Zerega et al., 2019*). Research permits were approved by the Ibu Pejabat Jabatan Hutan Wisma Sumber Alam (National Parks and Nature Reserves, Forest Department), Sarawak for 2016 (Permit no. NCCD.907.4.4(JLD.13)-195) and renewed for 2017 (Permit no. NPW.907.4.4(JLD.14)-137). All known sites from which *A. annulatus* had previously been recorded (Gunung Gayu, Gunung Mentawa, Gunung Teng Bukap, and Gunung Payang), as well as four additional sites in surrounding areas of limestone hill forest (Gunung Manok, Gunung Bedoh, Gunung Jambusan, and Fairy Caves) were visited to search for *A. annulatus* individuals (*Zerega et al., 2019*). Individuals of *A. annulatus* were found in five sites (Gunung Mentawa, Gunung Gayu, Gunung Teng Bukap, Gunung Manok, and Gunung Bedoh) (Fig. 2). Representative collections were made at each site, and the number and age class of individuals from each population varied (Table S1). At one of the sites where collections had previously been made in 1999 (Gunung Payang), the area was found in 2017 to have a small area of limestone hill forest remaining, and a nearby rubber plantation. However, no *A. annulatus* was found (pers. obs. Zakaria). For all sites where *A. annulatus* was found, GPS coordinates, images, and herbarium collections were taken for representative samples. In addition, leaf tissue samples (dried in silica), for 131 *A annulatus* trees were taken across the five sites (Table S1). For most of these 131 trees, diameter at breast height (dbh) measurements, and notes about the age class of the samples were also taken. The trees sampled were of varying maturity (seedling, sapling, and reproductively mature). Seedlings were defined as having a height less than 1 meter, while mature plants were defined as having a dbh of 5.5 cm or more, because plants that met this criterion showed signs of past reproduction. Anything taller than 1 meter and with a dbh less than 5.5 cm was considered a sapling. The dried leaf material was shipped to the Chicago Botanic Garden where it was stored at −20 °C. Additional data for comparison to closely related congeners of *A. annulatus* came from jackfruit (*A. heterophyllus, Witherup et al., 2019*, $n = 373$ from Bangladesh; and *Melhem, 2015*, $n = 373$ from India), cempedak (*A. integer* var. *integer*, $n = 344$), and *A. integer* var. *silvestris* ($n = 187$, bangkong, a wild relative of cempedak) (*Wang et al., 2018*).

## Genetic data collection

Genomic DNA was extracted from 1 cm$^2$ dried leaf using either the Qiagen DNeasy kits (Hilden, Germany) following standard protocol or a modified cetyltrimethylammonium

bromide (CTAB) method (*Doyle & Doyle, 1987*). Nine different microsatellite primers developed for *A. altilis* (*Witherup et al., 2013*) were used following the methods of *Witherup et al. (2013)*. These loci previously revealed diversity in the two most closely relatives of *A. annulatus*: jackfruit (*A. heterophyllus*) and cempedak (*A. integer*) (*Wang et al., 2018*; *Witherup et al., 2019*). The microsatellite regions amplified in this study were all dinucleotide repeats: MAA26 (black D2 dye), MAA54 (blue D4 dye), MAA105 (black D2 dye), MAA122 (green D3 dye), MAA140 (blue D4 dye), MAA156 (green D3 dye), MAA182 (blue D4 dye), MAA178 (green D3 dye), and MAA196 (black D2 dye). The forward primers were labelled with WellRed fluorescent dyes, shown in parentheses above.

The PCR reactions contained 5 µL of 2x myTaq Mix (Bioline, Taunton, Massachusetts, USA), 0.15 µL 20 mg/ml BSA (bovine serum albumin), 0.25 µL each of labelled forward and reverse primers (10 µm concentration), 3.35 µL of DNA-free water, and 1 µL of DNA (with concentrations between 10 and 80 ng/µL). The PCR cycles were: 2 min at 94 °C for initial denaturization, followed by 34 cycles of 94 °C for 30 s of denaturation, 52 °C for 1 min of annealing, then 72 °C for 2 min of extension; the 34 cycles were then followed by 72 °C for 10 min of final extension (*Wang et al., 2018*).

The PCR products were run on a Beckman Coulter CEQ 8000 Genetic Analysis System (Beckman Coulter, Brea, CA) following the methods of *Witherup et al. (2013)* and multiplexed via the same groupings of *Wang et al. (2018)*. The samples were scored alongside a GenomeLab 400 bp internal size standard ladder that was added to HiDi formamide (20 µL ladder per 1.5 mL of HiDi) (Azco Biotech., San Diego, CA, USA). In each well, 30 µL of the HiDi /ladder mixture were added to between 1.0 and 2.0 µL of PCR product, depending on the strength of the fluorescence color. These concentrations were determined by running a few different combinations and dilutions of primers and determining which resulted in the clearest peaks.

The resulting peaks were scored manually using Beckman Coulter CEQ 8000 Software 9.0. Allele sizes are recorded based on the size of the ladder and electrophoresis rates as described in *Wang et al. (2018)*. *Artocarpus annulatus* is thought to be functionally diploid, and as such each region should display either one (homozygous) or two (heterozygous) alleles. Previous studies have revealed that some primers (including primers MAA196 and MAA156) may amplify two regions, and up to four alleles can be present (*Wang et al., 2018*; *Witherup et al., 2013*; *Witherup et al., 2019*). This was rarely observed in *A. annulatus.* In a few samples when more than three strong peaks were observed, the rarest allele was omitted (allele size 268 for primer MAA196; allele size 170 for primer MAA54) (*Wang et al., 2018*).

### Analysis of genetic data

Microchecker analysis (*Van Oosterhout et al., 2004*), using the Oosterhout method was run to check for null alleles. The microsatellite data were analyzed using GenAlEx v.6.5 (*Peakall & Smouse, 2012*) to calculate diversity measures, including average number of alleles per locus ($N_a$), number of effective alleles ($N_e$), observed and expected heterozygosities ($H_o$ and $H_e$), number of private alleles ($P_a$), inbreeding coefficient ($F_{IS}$) and

fixation index ($F_{ST}$). Allelic richness was calculated in the R package Hierfstat (*Goudet, 2005*). To test if diversity measures were significantly different across populations and age classes, single factor ANOVAs were conducted in Excel v. 16.16.14.

Genetic structure among populations was visualized using STRUCTURE v.2.3.4 (*Pritchard, Stephens & Donnelly, 2000*). Twenty independent runs per $K$ were carried out with a burn-in period of 10,000 and 10,000 MCMC iterations for $K = 1$ –20. Structure Harvester (*Earl & VonHoldt, 2012*) was used to determine the most likely value of genetic clusters ($K$) implementing the Evanno method (*Evanno, Regnaut & Goudet, 2005*). Spatial clustering of populations was assessed using Principle Coordinate Analysis (PCoA) of Nei's genetic distance in GeneAlEx (*Peakall & Smouse, 2012*). Additionally, GEneAlEx v.6.5 (*Peakall & Smouse, 2012*) was used to assess genetic differentiation among populations AMOVA. To assess whether there was any genetic differentiation among age classes, PCoA, $F_{ST}$, and AMOVA analyses were also run for the three age classes in GeneAlEx v.6.5 (*Peakall & Smouse, 2012*). For PCoA, the two principle coordinates were plotted with 95% confidence ellipse in R (ggplot) (*Wickham, 2016*).

## RESULTS

### Microsatellite loci

Of the nine different microsatellite primers that were tested, one (MAA178) consistently amplified two readily distinguishable regions, which were scored as separate loci and differentiated as primer MAA178A and primer MAA178B (following *Wang et al., 2018*). Across the resulting 10 loci, there were 44 detected alleles, however, two loci were excluded from further analyses as they were monomorphic across all samples (MAA122 and MAA140) (Table S2). Another locus (MAA54) was excluded from further analyses as it was functionally monomorphic, in that nearly all individuals were heterozygotes with the same two alleles, suggesting that this primer may amplify two fixed homozygote loci (MAA54 has been shown to amplify two loci in other *Artocarpus* species) (*Wang et al., 2018*). Although these three monomorphic loci were excluded from subsequent analyses, the fact that the same loci were found to be polymorphic (with 3–18 alleles) in the two closest relatives to *A. annulatus* (*Wang et al., 2018*; *Witherup et al., 2019*), suggests that *A. annulauts* is more depauperate in genetic diversity. Finally, MAA105 was excluded from further analysis due to a null allele frequency value >0.20, which can cause significant underestimation of population differentiation (Table S2). This threshold value has been used in other studies to exclude primers that may significantly affect heterozygosity measures (*Chapuis & Estoup, 2007*; *Minn, Prinz & Finkeldey, 2014*; *Muzzalupo, Vendramin & Chiappetta, 2014*). The following loci were included in subsequent genetic diversity and structure analyses: MAA26, MAA156, MAA182, MAA178 A, MAA178 B, MAA196.

### Genetic diversity and structure

Analysis of the polymorphic loci with no evidence of null alleles revealed no significant differences in diversity measures across populations (Table 1). The Fixation Index of individuals ($F_{IS}$) for all populations was negative (indicating no signs of inbreeding), or in the case of Gunung Mentawa it was barely positive (0.05). The population at Gunung

**Table 1 Genetic diversity measures of *A. annulatus* populations.** Number of individuals sampled in each population by size class. Numbers in parentheses after population names refer to number of individuals from each size class (mat, mature; sap, sapling; sdlg, seedling).

| Population | N | $N_a$ | $N_e$ | $A_R$ | $H_o$ | $H_e$ | $F_{IS}$ | I | $P_a$ |
|---|---|---|---|---|---|---|---|---|---|
| Gunung Gayu (5 mat, 16 sap, 1sdlg) | 22 | 3.33 | 2.01 | 3.44 | 0.60 | 0.49 | −0.23 | 0.85 | 5 |
| Gunung Bedoh (11 mat, 12 sap, 3sdlg) | 26 | 3.83 | 1.81 | 3.14 | 0.49 | 0.41 | −0.20 | 0.72 | 6 |
| Gunung Manok (13 mat, 18 sap) | 21 | 2.00 | 1.50 | 2.53 | 0.42 | 0.29 | −0.36 | 0.45 | 1 |
| Gunung Mentawa (6 mat, 3 sap, 6sdlg) | 15 | 2.17 | 1.56 | 2.96 | 0.31 | 0.29 | 0.05 | 0.48 | 0 |
| Gunung Teng Bukap (6 mat, 20 sap, 21sdlg) | 47 | 3.00 | 1.71 | 3.08 | 0.38 | 0.35 | −0.06 | 0.63 | 3 |
| Total (41 mat, 59, sap, 31 sdlg) | 131 | | | | | | | | |
| P value | NA | 0.07 | 0.31 | 0.18 | 0.46 | 0.40 | 0.40 | 0.26 | 0.20 |

Notes.

$N_a$, no. of different alleles; $N_e$, no. of effective alleles; $A_R$, allelic richness; $H_o$, observed heterozygosity; $H_e$, expected heterozygosity; $F_{IS}$, Fixation Index; $P_a$, number of private alleles; I, Shannon's Informational Index.

**Table 2 Pairwise comparisons of $F_{ST}$ values.** Comparison of *A. annulatus* by population and tree size.

| | | Pairwise $F_{ST}$ Comparison by Populations | | | |
|---|---|---|---|---|---|
| Gunung Gayu (Pop. 1) | Gunung Bedoh (Pop. 2) | Gunung Manok (Pop. 3) | Gunung Mentawa (Pop. 4) | Gunung Teng Bukap (Pop. 5) | |
| 0.00 | – | – | – | – | 1 |
| 0.067 | 0.000 | – | – | – | 2 |
| 0.118 | 0.032 | 0.000 | – | – | 3 |
| 0.101 | 0.052 | 0.068 | 0.000 | – | 4 |
| 0.061 | 0.032 | 0.059 | 0.028 | 0.000 | 5 |

| | Pairwise $F_{ST}$ Comparison by Tree Size | | |
|---|---|---|---|
| Sapling | Seedling | Mature | |
| 0.00 | – | – | Sapling |
| 0.021 | 0.000 | – | Seedlling |
| 0.020 | 0.014 | 0.000 | Mature |

Mentawa had no private alleles, while the other four populations had between one (Gunung Manok) and six (Gunung Bedoh) private alleles. Pairwise comparisons of $F_{ST}$ showed limited levels of genetic differentiation across populations (Table 2). Results of AMOVA among and within populations showed that only 8% of diversity can be accounted for among populations, and 92% is within populations (Table 3).

All samples were sorted into age classes as described in the methods. Analysis of the six microsatellite loci revealed no significant differences in diversity measures across size classes, except in the case of private alleles; seedlings had significantly fewer private alleles than saplings or mature trees (Table 4). Pairwise comparisons of $F_{ST}$ across tree size class showed no signs of genetic differentiation across size classes (Table 2). Results of AMOVA among and within tree size classes indicated that only 6% of diversity can be accounted for among populations, and 94% is within populations (Table 3).

The first two principle coordinates of the PCoA account for 25% and 10%, respectively, of the variability in the dataset, and there are no clear clusters based on populations or tree size class (Fig. 3). The population at Gunung Mentawa occupies the smallest space,

**Table 3  AMOVA.** Results are shown among and within five *A. annulatus* populations, and among and within tree size classes across all five populations.

| Source | df | SS | MS | Est. Var. | % |
|---|---|---|---|---|---|
| | | AMOVA across five populations | | | |
| Among populations | 4 | 42.80 | 10.70 | 0.17 | 8 |
| Within populations | 257 | 534.27 | 2.08 | 2.08 | 92 |
| Total | 261 | 577.07 | – | 2.25 | 100 |
| | | AMOVA across three tree size classes | | | |
| Among populations | 2 | 28.15 | 14.08 | 0.14 | 6 |
| Within populations | 259 | 548.92 | 2.12 | 2.12 | 94 |
| Total | 261 | 577.07 | | 2.26 | 100 |

Notes.

df, degrees of freedom; SS, sum of squares; MS, mean square; Est. Var., estimated variance.

**Table 4  Genetic diversity measures by tree size class of *A. annulatus* across five populations.** Total values are listed for sample size (N) and private alleles ($P_a$). Mean values across all loci are listed for all other measures. If *p*-value are $< 0.05$, different superscript letters indicate significant difference between groups.

| Size Class | N | $N_a$ | $N_e$ | $A_R$ | $H_o$ | $H_e$ | $F_{is}$ | I | $P_a$ |
|---|---|---|---|---|---|---|---|---|---|
| Seedling | 28 | 3.00 | 1.77 | 3.46 | 0.41 | 0.36 | −0.16 | 0.65 | 0[a] |
| Sapling | 46 | 4.33 | 1.80 | 4.02 | 0.45 | 0.42 | −0.06 | 0.77 | 8[b] |
| Mature | 37 | 4.00 | 1.73 | 3.30 | 0.43 | 0.39 | −0.10 | 0.70 | 6[b] |
| *P* value | NA | 0.26 | 0.97 | 0.12 | 0.95 | 0.89 | 0.79 | 0.84 | **0.03** |

Notes.

$N_a$, no. of different alleles; $N_e$, no. of effective alleles; $A_R$, allelic richness; $H_o$, observed heterozygosity; $H_e$, expected heterozygosity; $F_{IS}$, Fixation Index; $P_a$, number of private alleles; I, Shannon's Informational Index.

as does the mature tree size class. STRUCTURE was run on all samples with six loci, and results were analyzed using the Evanno method in Structure Harvester (*Earl & VonHoldt, 2012*) to determine the most probable number of genetic clusters ($K$). The most likely number was $K = 3$. The structure plot shows that the genetic clusters are restricted to neither distinct geographic populations nor size classes of trees, with all three clusters being fairly equally present in each population and each size class (Fig. S1).

## Genetic diversity comparisons with closely related species

Expected heterozygosities ($H_e$) and inbreeding coefficients ($F_{IS}$), based on the same six microsatellite loci, were calculated for *A. annulatus* and its closest relatives: *A. integer* var. *integer* (cempedak),*A. integer* var. *silvestris* (bangkong, a wild relative of cempedak) (Data from *Wang et al., 2018*), and *A. heterophyllus* (jackfruit) (Data from *Witherup et al., 2019*). A one-way ANOVA found no significant difference ($p = 0.113$) across the species for $H_e$, though *A. annulatus* tended to have the lowest values (Table 5). A one-way ANOVA did find significant difference ($p = 0.020$) across species for $F_{IS}$, and a posthoc Tukey HSD test then found significant differences between *A. annulatus* and the other three taxa (*A. heterophyllus*, $p < 0.01$; *A. integer* var. *integer*, $p = 0.013$; *A. integer* var. *silvestris*, $p = 0.039$)

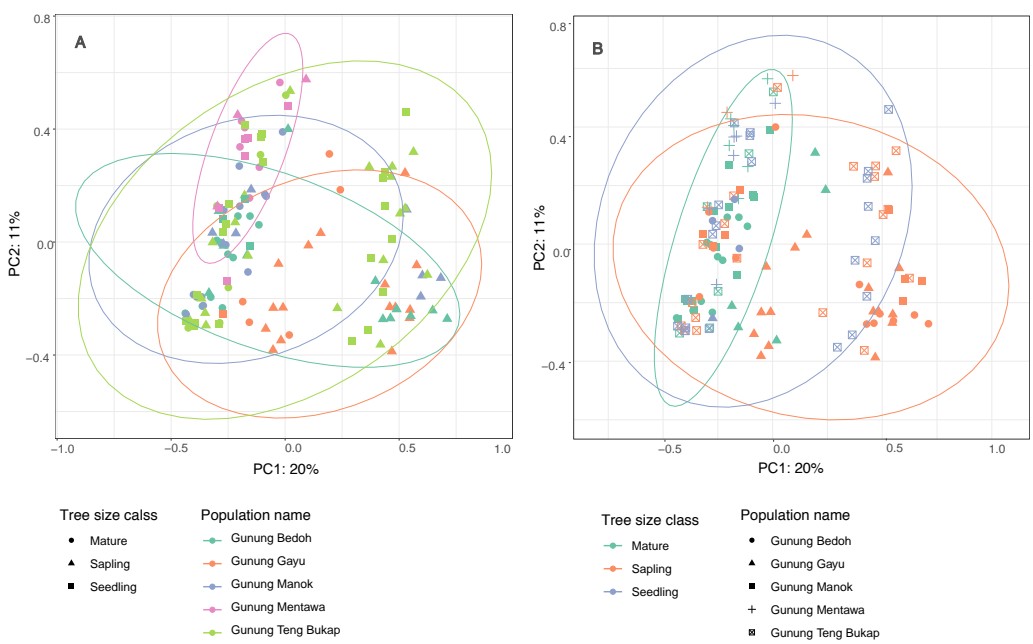

**Figure 3  Principle Coordinate Analysis of five *A. annulatus* populations.** (A) Populations are displayed in different colors, tree size class is represented by different shapes, 95% confidence ellipses are drawn around populations; (B) tree size classes are displayed in different colors; populations are represented by different shapes; 95% confidence ellipses are drawn around tree size classes.

**Table 5  Comparison of genetic diversity ($H_e$).** Comparisons are shown between *A. annulatus* and its closest relatives, *A. heterophyllus* (jackfruit) and *A. integer* (cempedak) based on the same microsatellite loci. There is no significant difference between species for $H_e$, but there is for $F_{IS}$ (indicated by superscript letters).

| Locus | *A. heterophyllus* (N = 746) | | *A. integer* var. *integer* (N = 344) | | *A. integer* var. *silvestris* (N = 187) | | *A. annulatus* (N = 131) | |
|---|---|---|---|---|---|---|---|---|
| | $H_e$ | $F_{IS}$ | $H_e$ | $F_{IS}$ | $H_e$ | $F_{IS}$ | $H_e$ | $F_{IS}$ |
| MAA26 | 0.774 | 0.106 | 0.348 | 0.341 | 0.480 | 0.154 | 0.427 | −0.094 |
| MAA156 | 0.373 | −0.084 | 0.566 | 0.016 | 0.613 | −0.088 | 0.287 | −0.429 |
| MAA182 | 0.554 | 0.223 | 0.178 | 0.360 | 0.594 | 0.182 | 0.087 | −0.146 |
| MAA178A | 0.580 | 0.638 | 0.610 | 0.092 | 0.553 | 0.261 | 0.484 | 0.091 |
| MAA178B | 0.776 | 0.391 | 0.558 | 0.255 | 0.870 | 0.207 | 0.535 | −0.105 |
| MAA196 | 0.730 | 0.609 | 0.599 | 0.165 | 0.521 | 0.179 | 0.497 | −0.439 |
| **Mean** | **0.631** | **0.314[a]** | **0.477** | **0.205[a]** | **0.605** | **0.149[a]** | **0.386** | **−0.184[b]** |

## DISCUSSION

### Genetic diversity of closely related *Artocarpus* species

Understanding genetic diversity and its spatial and temporal structure within endangered species and populations is important in order to make sound conservation management plans, especially for imperiled species with very restricted habitats. When starting with zero information about the population genetics of a species, assessing standard measures

of genetic diversity is important to establish a baseline of information, but it is also valuable to consider those measures in a broader context. Looking at closely related, more common congeners allows for an evolutionary, comparative perspective. For example, *Widener & Fant (2018)* found higher levels of inbreeding in an endangered edaphic specialist compared to its more common congener. In the case of *A. annulatus* and its closest relatives, when considering measures of diversity ($H_e$) of its most closely related congeners (*A. heterophyllus, A. integer* var. *integer*, and *A. integer* var. *silvestris*) using the same six microsatellite markers, *A. annulatus* tends to have lower measures, but there is no significant difference ($p = 0.597$) (Table 5). We also considered broader comparisons with other long-lived, outcrossing wild trees and closely related domesticates. *Miller & Gross (2011)* reviewed genetic diversity in perennial crops and their wild relatives and reported that based on neutral codominant markers (SSRs and allozymes) cultivated perennial fruit crops retain on average 95% (from 65%–127%) of the neutral variation found in wild populations. Cultivated *Artocarpus* crops had elevated levels of genetic variation compared to the closely related *A. annulatus* (Table 5). This could be an artifact of smaller sampling size in *A. annulatus*, or it could reflect loss of diversity due to habitat destruction. Another possible explanation is that the cultivated populations may represent descendants of crosses between geographically and genetically distinct individuals, giving rise to novel diversity. Studies have demonstrated that allelic diversity in small populations are likely to be very low simply by chance (*Cole, 2003*), which may be the case for *A. annulatus*.

Despite the lower than expected levels of diversity, *A. annulatus* does not show signs of inbreeding, as evidenced by the negative value of the $F_{IS}$ compared to its three relatives—displaying $F_{IS}$ values between 0.149 and 0.314 (Table 5). The results might seem counter-intuitive for such small and restricted populations. However, since most of the variation present in *A. annulatus* is distributed within populations (93%), and because *A. annulatus* is a long-lived, likely outcrossing tree (as are other members of the genus), these results are put into perspective. Genetic diversity levels could be remnant from when the population of *A. annulatus* may have been larger and may have had higher levels of diversity, and the remaining individuals still retain this because of their lifespan (*Guries & Ledig, 1979*). Additionally, it is possible that *A. annulatus* shows no signs of inbreeding, because long-lived perennials often exhibit stronger selection than non-perennials against inbred plants. Indeed, most woody perennials exhibit high outcrossing rates to maintain greater levels of diversity (*Duminil, Hardy & Petit, 2009*). Meanwhile, crops like jackfruit (*A. heterophyllus*) are under human selection and are frequently vegetatively propagated (i.e., grafting) to maintain desired traits (*Witherup et al., 2019*).

## Genetic structure within and across *Artocarpus annulatus* populations

The results of several analyses indicate that there is little to no geographical genetic structure exhibited across the five populations of *A. annulatus*. Principle Coordinate Analysis and Structure analyses indicate there is no genetic differentiation based on the geography of the populations (Fig. 3, Fig. S1). The results from the AMOVA test revealed that only 8% of diversity was found among populations (Table 3), and the

$F_{ST}$ values demonstrated low genetic differentiation among populations (Table 2). Values below 0.05 are considered to represent very low to no genetic differentiation (*Frankham, Ballou & Briscoe, 2002*), and the values in this study ranged from 0.028 to 0.089. The five populations have an area of occupancy of approximately 32 km$^2$ (*Zerega et al., 2019*). Three of the populations (Gunung Gayu, Gunung Mentawa, and Gunung Teng Bukap) are geographically closely clustered, each being ca. 0.3 km apart. Gunung Manok is the most isolated, situated ca. 6 km from the cluster; and Gunung Bedoh is in the middle, ca. 3 km away from both Gunung Manok and the cluster of three (Fig. 2). Not surprisingly, Gunung Manok is the most differentiated from the others, being the least differentiated from the closest population (Gunung Bedoh). These results, taken in aggregate, suggest that gene flow has occurred among the populations, but that it may be limited across greater distances. The five *A. annulatus* populations may be part of one largely intermixing population across which genetic material is shared via gene flow or may represent remnants of populations that were once larger and less fragmented. Because it is a long-lived tree, the effects of fragmentation and isolation may be slower manifested in the genetic structure.

It is of interest to consider methods of gene flow to understand what might be causing low levels of genetic differentiation across populations in such a dramatic landscape as karst forests, where dispersal ability of inhabitants may be limited (*Hughes, 2017*). The two ways for plants to spread genes across space are via pollen dispersal (pollination) and fruit/seed dispersal. While we know virtually nothing about this in *A. annulatus*, we can turn to its closest relatives to consider possibilities. Recent work has been conducted on elucidating pollination mechanisms in its two closet relatives, jackfruit (*A. heterophyllus*) and cempedak (*A. integer*). Both of these monoecious species (separate male and female inflorescences present on the same individual tree) are thought to be pollinated via a three-way mutualistic relationship between the plant itself, gall midges (genus: *Clinodiplosis*, Family: Cecidomyiidae, Order: Diptera), and a fungus (yet to be identified, but possibly Genus: *Choanephora*, Family: Choanephoraceae, Order: Mucorales) that grows on the male inflorescences (*Gardner et al., 2018*; *Sakai, Kato & Nagamasu, 2000*). The adult gall midges are thought to be attracted to the same volatile compounds that are similarly emitted by both male and female inflorescences. The midges feed on the pollen of male flowers, and at the same time a fungus (which does not appear to infest other parts of the tree) gradually grows over the length of the male inflorescence. Gravid female midges lay their eggs on the fungus and the young larvae feed on the liquid exuded by the fungal mycelia. As the midges seek out male inflorescences for feeding and egg-laying, they sometimes mistakenly visit female flowers (which offer no apparent reward for the midges but mimic the scent of the male flowers) (*Gardner et al., 2018*), and in the process may transfer pollen to effect pollination. Given that its closest relatives are both pollinated by gall midges, it is hypothesized that *A. annulatus* may share a similar mechanism (*Gardner et al., 2018*). Gall midges are tiny insects, about 1 mm in length, and have short lifespans (adults live between a few hours to a few days), so they are likely limited in the distances they can travel to disperse pollen (*Gardner et al., 2018*; *Skuhrava, 1991*). Some of the *A. annulatus* populations in this study are found within just 0.25 km of each

other, while the most isolated population (Gunung Manok) is about 3 km to the closet population (Gunung Bedoh) and about 6 km to the next closet populations. Most of what is known about flight range of members of the gall midge family comes from studies of pests. For example, studies of Hessian flies (*Mayetiola destructor, Cecidomyiidae*) suggest that they choose when, where, and how far to move based on cues from the environment, and are also affected by wind (*Schmid et al., 2018*). They can disperse from a few meters to a few kilometers and exhibit nonrandom movement directed by things like scents or certain wavelengths of color (*Harris & Rose, 1990*; *Withers, Harris & Madie, 1997*; *Anderson et al., 2012*). Additionally, female midges are more likely to stay in an area where their plant of interest is located as opposed to moving to an area with less attractive plants (*Withers & Harris, 1996*). We still need to learn more about pollination in *A. annulatus*, but it seems possible that if it is similar to its closest relatives, gall midges do not typically travel long distances, and the more fragmented *A. annulatus* populations become over time, the more greatly genetically differentiated they could become. This may explain why the most geographically distant population (Gunung Manok) is also the most genetically differentiated from the others.

However, none of the populations are very highly differentiated from one another, and gene flow appears to be occurring (or has recently occurred) among them (Table 2, Fig. S1). This brings us to the other mechanism for gene flow: fruit (i.e., seed) dispersal. The fruits of *A. annulatus* are large (ca. 6 × 8 cm) and distinctive with bumpy or spiky skin and a strong fragrance (*Jarrett, 1975*). Just like its crop relatives, jackfruit and cempedak, the fruit structure grows from the trunk of the tree (Fig. 1), but the wild *A. annulatus* has much smaller and less fleshy fruits than its crop relatives (*Jarrett, 1975*). While research is limited on fruit dispersal in *Artocarpus*, there are studies that suggest fruits are spread by large mammals, and this may also be the case in *A. annulatus*. In India, the related *A. chaplasha* depends on large mammals such as the Asian elephant (*Elephas maximus*), domestic cows (*Bos primigenius*), buffalo (*Bubalus bubalis*), and rhesus macaques (*Macaca mulatta*) as seed dispersers (*Sekar, 2014*). The study species, *A. annulatus,* shares its habitat with a few sizeable mammals, including François's leaf monkey (*Trachypithecus francoisi*) and the serow (*Capricornis sumatraensis*) (*Kiew et al., 2017*). While anecdotal information from fieldwork suggests that many of the seeds simply germinate under the mother tree, where the fruits drop (i.e., numerous saplings were observed growing in clusters beneath mature trees), it is possible that birds or mammals disperse some *A. annulatus* seeds, accounting for gene flow and low genetic differentiation across sites (*Zerega et al., 2019*). Future research could focus on better understanding the contributions of these two modes of gene flow by studying the distribution of genetic structure of seed (maternal chloroplast loci) vs. pollen dispersed alleles.

## Genetic structure across age classes of *Artocarpus annulatus*

When an endangered species occurs in habitats that are under threat, it is valuable to know if genetic diversity is being lost over time, such that older, mature individuals harbor greater or unique genetic diversity compared to young seedlings and saplings. As the highly specialized karst habitat becomes increasingly smaller and fragmented due to

human pressures, it might be expected that the *A. annulatus* gene pool could be reduced due to random, stochastic loss of individuals, leading to a decreased level of diversity in younger generations and a genetic bottleneck. When *A. annulatus* individuals were analyzed by size class (seedling, sapling, or reproductively mature), diversity measures tended to be lower in seedlings compared to saplings and mature trees. However, only the number of private alleles was significantly less in seedlings (Table 4). Additionally, PCoA clustering analysis shows that the age classes do not cluster separately (Fig. 3), and AMOVA (Table 3) suggests that only 6% of diversity was found within age classes. This all suggests that there is no genetic differentiation across size classes, but that allelic diversity may be under threat in the youngest generation of plants. To better understand possible temporal changes, it would be useful to look at a much longer trajectory of time and over a larger space. Assessing the genetic diversity of older herbarium specimens would be one way to help determine if diversity has been lost. Unfortunately, there are only nine historical collections known of *A. annulatus,* the oldest only dates back to the 1960s, and due to its critically endangered status, sampling of herbarium specimens for DNA was not possible. However, the present study provides a baseline estimate of genetic diversity that can be used for comparison of future studies.

### Informing conservation strategies

Understanding *A. annulatus* genetic diversity can inform conservation management, including ex situ conservation. In the case of *A. annulatus,* different populations were not all strongly genetically differentiated from one another, so there may be little justification for collecting germplasm from multiple individuals from all populations for ex situ conservation in local arboreta and conservation collections. Instead, seeds or seedlings from populations with the most individuals or the most accessible population could be targeted for conservation. For in situ conservation efforts, all sites should ideally be protected (*Crozier, 1997*), however, focusing conservation efforts on the largest population would likely yield similar results in terms of conserving the breadth of *A. annulatus* genetic diversity. In many cases, in situ conservation may be the preferred approach. A recent meta-analysis of 32 plant species from various habitats compared how well plants grew in their local habitats versus a foreign transplant site. They found that in over 70% of the studies the local plants consistently did better in their habitats of origin, further supporting the argument for conservation of karst areas (*Leimu et al., 2006*). Additionally, the unique karst habitat is difficult to replicate, and due to the economic importance and biodiversity that karsts hold, in situ conservation is especially important as it conserves an entire ecosystem.

## CONCLUSIONS

The threats to limestone karst ecoystems in Southeast Asia are immense, and we still have much to learn about them (*Clements et al., 2006*; *Hughes, 2017*). Laws to protect karst ecosystems are severely lacking, and policies in place are often not enforced (*Clements et al., 2006*). Unfortunately, the lack of information about the ecosystems and the biodiversity they house can make it more difficult to justify their conservation. For

this reason, it is important to conduct research in these ecosystems, and by combining available data across studies it can help understand patterns and prioritize regions for protection. *Artocarpus annulatus* is just one of many endemic species in an ecosystem that is as incredibly unique and diverse as it is fragile, and points to the need for protection of the karst forests of Southeast Asia. The health of *A. annulatus*, as a long-lived, fruit-bearing tree, is crucial for the complex web of plants and animals that depend upon it. Finally, given that *A. annulatus* is the closest wild relative of two important fruit tree species, it may be valuable for breeding programs to expand the growing conditions of jackfruit and cempedak to limestone soils.

## ACKNOWLEDGEMENTS

We would like to thank Jeremie Fant, Fabrizio Grassi, and two anonymous reviewers for valuable and insightful comments that improved this manuscript; Lynnaun Johnson for assistance in the lab; Jegong anak Suka and Jugah anak Tagi for assistance in the field; and the following herbaria for access to collections: SAR, KEP, L, K, HUH.

### Funding

This research was supported by Northwestern University's Undergraduate Research Grant program, the Mohamed bin Zayed Species Conservation Fund (project number 152511619), and the National Science Foundation (DBI award number 1711391). The funders had no role in study design, data collection and analysis, decision to publish, or preparation of the manuscript.

### Grant Disclosures

The following grant information was disclosed by the authors:
Northwestern University's Undergraduate Research Grant program, the Mohamed bin Zayed Species Conservation Fund: 152511619.
National Science Foundation: 1711391.

### Competing Interests

The authors declare there are no competing interests.

### Author Contributions

- Leta Dickinson performed the experiments, analyzed the data, prepared figures and/or tables, authored or reviewed drafts of the paper, and approved the final draft.
- Hilary Noble performed the experiments, authored or reviewed drafts of the paper, and approved the final draft.
- Elliot Gardner conceived and designed the experiments, authored or reviewed drafts of the paper, conducted fieldwork, and approved the final draft.
- Aida Shafreena Ahmad Puad and Wan Nuur Fatiha Wan Zakaria conceived and designed the experiments, authored or reviewed drafts of the paper, conducted fieldwork, and approved the final draft.

- Nyree J.C. Zerega conceived and designed the experiments, performed the experiments, analyzed the data, prepared figures and/or tables, authored or reviewed drafts of the paper, conducted fieldwork, and approved the final draft.

**Field Study Permissions**

The following information was supplied relating to field study approvals (i.e., approving body and any reference numbers):

Ibu Pejabat Jabatan Hutan Wisma Sumber Alam (National Parks and Nature Reserves, Forest Department), Sarawak approved fieldwork conducted in 2016 and 2017. For 2016 fieldwork: Permit no. NCCD.907.4.4(JLD.13)-195. For 2017 fieldwork: renewed as permit no. NPW.907.4.4(JLD.14)-137.

**Data Availability**

Allele calls for microsatellite data are available in a Supplemental File.

**Supplemental Information**

Supplemental information for this article can be found online at http://dx.doi.org/10.7717/peerj.9897#supplemental-information.

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
