# Peer review of "Genetic diversity and structure of the critically endangered Artocarpus annulatus, a crop wild relative of jackfruit (A. heterophyllus)"

_PeerJ, doi:10.7717/peerj.9897_

## Round 0.1 · original submission · Major Revisions

Dear Dr. Zerega,

The reviewers think your research is interesting and that it has some potential and valuable data. Moreover, they find the manuscript generally well written. Nevertheless, they have some concerns and think that the scientific soundness of the study has to be improve.

So, I encourage you to improve the manuscript according to all tips of reviewers. Please, respond point-to-point to the comments of reviewers to speed up the process of revision.

Once again, thank you for submitting your manuscript to PeerJ and we look forward to receiving your revision.

Sincerely,
Gabriele Casazza

·

Basic reporting

Six microsatellite markers were used to assess genetic diversity in five populations of
Artocarpus annulatus. Moreover, leaf material, locality information, and demographic data were collected from them.

I agree that locate and conserve crop wild relatives is critical for the conservation of crop genetic diversity and improvement. Moreover, I acknowledge that studying and conserving A. annulatus has broad economic implications. On the other hand, I observe some important limitations in the experimental design.
As observed by authors, the results show the absence of inbreeding and genetic structure among the five populations in A. annulatus. However, this result could be afflicted by the low number of microsatellites analysed as observed for other wild species. In my opinion, the authors should increase the number of genetic markers to assess the real genetic diversity and structure among populations. I recognize that finding or developing new SSR markers is very demanding but this is the only way to get robust results. Alternatively, the authors should use other molecular tools such as AFLP or RAD-seq, which could allow collecting more data. Instead, I evaluate very positively to analyze the genetic structure across age classes. This result could be very interesting in conservation program.

Experimental design

Methods are described with sufficient detail but the main problem is the low number of SSR analysed.

Validity of the findings

'no comment'

Reviewer 2 ·

Basic reporting

The article basically deals with the characterization of the diversity and genetic structure of populations of A. annulatus. This species occurs in a very particular, fragmented and threatened environment. In addition to describing the genetic diversity of the species, the authors make comparisons with closely related species and assess whether there is a difference in the diversity indexes when comparing different age groups. In general the article is well written but needs to improve in some points. The comments and suggestions are directly in the attached pdf, but I highlight some here:

Major points

- It is not clear how the data presented in Table 6 were obtained. I did not find in the cited literature whether the authors actually used the same loci and how the indexes presented in Table 6 were obtained. This should be better clarified throughout the text. See also comments made in the pdf file.

- In the discussion, indices calculated in the article by Miller and Gross (2011) were cited, see lines 294-297. I think it is difficult to make a comparative analysis, in terms of values, considering the results obtained in the present study. I suggest rethinking these sentences.

- The discussion item “Genetic structure within and across Artocarpus annulatus populations” should be completely revised, as the results obtained do not seem to coincide with the discussion on this topic. See comments in the pdf file.

- In the discussion, it is understood that the species under study is monoecious, but this was not mentioned in the manuscript, although there are some suggestions. This should be better clarified and maybe discussed with more emphasis.

Minor points

- Check the correct way of writing all indexes. In heterozygosities, the H must be in italics, the F indices must also be in italics and the subscribed letters are capitalized. Also review K. See some indications in the text in pdf.

- The Figure 2 (map) should be improved. Is box C showing the geographical location of the sampled populations? This should be seen directly on the map.

- Please increase symbols and legends in figure 3 and improve resolution.

- In Table 2 what is N? and what is NA? This should be in the legend.

- In table 4, standardize: populations or pops?

Experimental design

The experimental design is adequate.

Validity of the findings

I think the results obtained are a good contribution to the study area and may help to outline conservation strategies for the species.

Additional comments

See more comments in the attached pdf.

Annotated reviews are not available for download in order to protect the identity of reviewers who chose to remain anonymous.

Reviewer 3 ·

Basic reporting

no comment

Experimental design

no comment

Validity of the findings

no comment

Additional comments

The reviewed article concerns genetic diversity of critically endangered species Artocarpus annulus as revealed by microsatellite markers. Authors used to microsatellite primers to asses of genetic variation of 5 populations of A. annulus and closely related species: cempedak and jackfruit. The study has some potential and valuable data but i I can't recommended the article for publication in present form. Below I presented my general suggestion, that could improve the scientific soundness of this study:

General comments:
Introduction:
The introduction requires corrections and supplementation of additional information. Details below.
1) In the introduction, authors describe the karst ecosystem, the characteristic features of this habitat and problems regarding the disappearance of this habitat. Of course this information are important and useful to understand the conservation of analysed population of A. annulus. However because the article is about genetic diversity, structure and conservation of A. annulus the introduction should be more focused towards conservation genetics. I suggested shortening the paragraphs about the kars ecosystems and pay more attention to studied species. Example there is a lack of important information about study species (concerning biology and population resources), which are necessary in the discussion about genetic variation and strategies of conservation. This information would be help to understand justification for performed analyses. On the other hand, description of karst ecosystem too extensive for this purpose.
2) In the introduction, there is no scientific hypothesis, which should be assumed in this kind of study? Does authors suspect the genetic diversity as a bottleneck of population fitness?
3) It is also unclear how the genetic analysis of this species can impact management plans for in situ and ex situ conservation efforts …..(line 113-115). It should be explained.
Material and methods.

1) Microsatellite methods.

In this research authors used microsatellite markers which were developed for A. altilis (Witherup et. al. 2013) and tested for closely related species like cempedak and jackfruit (Wang et. al. 2018). Moreover for cempedak were observed more than two strong peaks and the rarest and weakest peaks were omitted. Additional peaks were also observed in this studies (lines 203-205 in manuscript). In addition, the starter MAA 178 amplified two readily distinguishable regions and authors was considered as two separate loci. The author should better explain why this two peaks were treated as two separate loci. In my opinion such loci should be excluded from the analysis and SSR primer specific for A. annulus should be developed, as it’s fast and cheap nowadays by using NGS sequencing.


2) Results

Genetic diversity and genetic structure.

1. I suggest to merge the tables 1 and 2.

2. I suggest counting the allelic richness and adding this information to table. Allelic richness is a straightforward measure of genetic diversity, that is commonly used in studies based on molecular markers that aim at selecting populations for conservation. This is very informative parameter, dependent on effective population size, a good indicator of past demographic changes that would have affected genes associated with adaptive traits as well as natural markers.

3. In the results, authors were repeated the information about results of Amova analysis. (line 250-252 and 257-259).

4. The Fig 3 is completely unreadable and should be corrected.
5. Structure analysis. I suggest to present lnP(D)/K in a simple graph. This method does also show a “peak” at K=12 which might be also worth to be presented and discussed.

---

## Round 0.2 · Minor Revisions

Dear Dr. Zerega,

The reviewers think your manuscript was strongly improved and suggest only minor changes. So, I ask you to perform these few changes before the manuscript's acceptance for publication. Please, respond point-to-point to the comments of reviewers to speed up the process of revision.
Once again, thank you for submitting your manuscript to PeerJ and we look forward to receiving your revision.
Sincerely,
Gabriele Casazza

·

Basic reporting

-

Experimental design

I am impressed by the authors' effort to improve the manuscript. The manuscript has improved in both introduction and discussion. On the other hand, I believe that the number of markers used for this purpose is too limited. Moreover, I have seen in Supplemental materials that SSR data have a large number of missing data.
This could affect the final result. I remain of the idea that authors should add more SSRs to improve the article. A sufficient number of markers would allow obtaining an excellent job.

Validity of the findings

-

Reviewer 2 ·

Basic reporting

No comment

Experimental design

No comment

Validity of the findings

No comment

Additional comments

The manuscript was carefully revised according to the reviewers' suggestions. I consider that all corrections and changes were made accordingly, making the article much stronger. Attached is the manuscript with minor corrections. After these corrections are made, I believe that it is ready for publication.

Annotated reviews are not available for download in order to protect the identity of reviewers who chose to remain anonymous.

Reviewer 3 ·

Basic reporting

no comment

Experimental design

no comment

Validity of the findings

no comments

Additional comments

Dear Authors, most of my comments have been included in this version of the manuscript. I have one more suggestion. Can the authors comment on the values of allelic richness in the Results and Discussion.

---

## Round 0.3 · accepted · Accept

Dear Dr. Zerega,

I am very pleased to say that your paper "Genetic diversity and structure of the critically endangered Artocarpus annulatus, a crop wild relative of jackfruit (A. heterophyllus)" is accepted for publication in PeerJ. Congratulations!

Thank you for submitting your work to PeerJ.

Sincerely,
Gabriele Casazza